# Synthesis and Characterization of Porous MgO Nanosheet-Modified Activated Carbon Fiber Felt for Fluoride Adsorption

**DOI:** 10.3390/nano13061082

**Published:** 2023-03-16

**Authors:** De-Cai Wang, Min-Da Xu, Zhen Jin, Yi-Fan Xiao, Yang Chao, Jie Li, Shao-Hua Chen, Yi Ding

**Affiliations:** 1Anhui Advanced Building Materials Engineering Laboratory, Anhui Jian Zhu University, Hefei 230601, China; 2School of Materials and Chemical Engineering, Anhui Jian Zhu University, Hefei 230601, China; 3School of Environment and Energy Engineering, Anhui Jian Zhu University, Hefei 230601, China

**Keywords:** porous MgO nanosheets, activated carbon fiber felt, modification, fluoride, adsorption

## Abstract

In the present work, the porous MgO nanosheet-modified activated carbon fiber felt (MgO@ACFF) was prepared for fluoride removal. The MgO@ACFF was characterized by XRD, SEM, TEM, EDS, TG, and BET. The fluoride adsorption performance of MgO@ACFF also has been investigated. The adsorption rate of the MgO@ACFF toward fluoride is fast; more than 90% of the fluoride ions can be adsorbed within 100 min, and the adsorption kinetics of MgO@ACFF can be fitted in a pseudo-second-order model. The adsorption isotherm of MgO@ACFF fitted well in the Freundlich model. Additionally, the fluoride adsorption capacity of MgO@ACFF is larger than 212.2 mg/g at neutral. In a wide pH range of 2–10, the MgO@ACFF can efficiently remove fluoride from water, which is meaningful for practical usage. The effect of co-existing anions on the fluoride removal efficiency of the MgO@ACFF also has been studied. Furthermore, the fluoride adsorption mechanism of the MgO@ACFF was studied by the FTIR and XPS, and the results reveal a hydroxyl and carbonate co-exchange mechanism. The column test of the MgO@ACFF also has been investigated; 505-bed volumes of 5 mg/L fluoride solution can be treated with effluent under 1.0 mg/L. It is believed that the MgO@ACFF is a potential candidate for a fluoride adsorbent.

## 1. Introduction

Excess fluoride in drinking water is a global problem. At low concentrations, fluoride-contained water is helpful in preventing tooth decay and dental caries [1,2]. However, when the concentration of fluoride exceeds 1.0 mg/L, it will cause dental fluorosis and bone fluorosis and affect the intellectual development of children [3]. Among various fluoride treatments, the adsorption method is valued for its excellent performance and economy. Nanoadsorbents with the advantage of a high surface area and more active sites always exhibit superior fluoride adsorption properties. However, due to the small size, it is difficult to fully recover the nano-adsorbents from water, which would cause secondary pollution to the environment. It is rational to load the nano-adsorbents on various substrates [4]; in this way, the large overall size of absorbents makes them easy to recover from water while maintaining high adsorption performances. 

Activated carbon, especially self-supporting activated carbon materials, such as activated carbon fiber felt (ACFF) with high surface area, abundant pore structure, and three-dimensional network structure has been widely studied as an excellent adsorbent [5,6,7]. However, the adsorption performance of activated carbon toward fluoride is very poor, which can be ascribed to the negative zeta potential at the neutral state of the activated carbon [8]. Modifying high-performance fluoride nano-adsorbents on the activated carbon substrates can efficiently improve their fluoride removal performance. For example, Esmeralda et al. modified lanthanum oxyhydroxides on the commercial granular activated carbon, and the fluoride adsorption capacity of the modified activated carbon was five times higher than that of the unmodified sample [9]. Pang et al. modified carbon fibers with zirconium nanomaterials, and the fluoride adsorption capacity of the adsorbent increased to 28.5 mg/g [2]. Although many reports focused on activated carbon-based fluoride adsorbents, the reports about the ACFF-based fluoride adsorbents are limited, and the fluoride adsorption of the ACFF-based adsorbent is still very low. Up to now, it is still a daunting challenge to obtain high-performance ACFF-based fluoride adsorbents.

Magnesium oxide (MgO), with the nature of strong affinity, low cost, and no toxicity, is widely studied for fluoride removal. Many reports have proven that MgO nanomaterials always exhibit excellent fluoride adsorption properties [10]. For example, Kong et al. found that the fluoride adsorption capacity of the MgO nanosheets reached 185.5 mg/g [11]. In order to overcome the drawback of the difficult recovery of MgO nanomaterials, several pioneering works have modified the MgO nanomaterials on activated carbon materials. Shahkarami et al. synthesized the MgO nanoparticle-modified activated carbon through the impregnation technique, and the loading amount of MgO nanoparticles was about 10% [12]. Using the spray technique, Siriwardane et al. [13] fabricated the MgO/activated carbon composite, and the content of MgO in the composite was about 1.2%. Although the MgO nanomaterial-modified activated carbon materials are often reported [14,15], the MgO nanomaterial-modified ACFF is rarely mentioned. Thus, it is urgent to obtain the MgO nanomaterial-modified ACFF composites with high loading amounts. In this work, the porous MgO nanosheet-modified activated carbon fiber felt (MgO@ACFF) with ultrahigh loading amount was obtained through a facile hydrothermal method. The loading amount of the porous MgO nanosheets reaches 810%, which is the largest loading amount concerned. The MgO@ACFF combines the advantages of network substrate and high-performance fluoride nano-adsorbent, which provides the merits of high adsorption capacity and easy isolation. The adsorption capacity of MgO@ACFF is larger than 212.2 mg/g. The adsorption mechanism of the MgO@ACFF composite also has been studied. It is believed that the MgO@ACFF is a promising candidate for fluoride removal.

## 2. Materials and Methods

### 2.1. Preparation

The activated carbon fiber felt was purchased from Anhui Jiahang Carbon Fiber Co., Ltd., Suzhou, Anhui, China. All of the chemical reagents (analytical grade) were purchased from (Sinopharm Chemical Reagent Co., Ltd., Shanghai, China), without further purification.

In a typical synthesis process, ACFF with appropriate size was put into NaOH solution and heated at 120 °C for pretreatment. Then the ACFF was washed with de-ion water several times and dried. A certain amount of MgSO_4_·7H_2_O and urea was added to 100 mL of deionized water and stirred vigorously for 15 min. Next, the ACFF was put into this transparent solution, sealed into a 100 mL autoclave, and heated at 120 °C for 12 h to obtain the precursor-modified ACFF. Finally, precursor-modified ACFF was put into a tube furnace and annealed under an N_2_ atmosphere at 500 °C for 2 h to obtain MgO@ACFF.

The loading amount of the modified MgO is calculated as follows:(1)$$loading\;amount=\frac{W_{MgO}}{W_{ACFF}}\times 100\%$$
where W_MgO_ is the weight of the modified MgO nanosheets, and W_ACFF_ is the weight of the ACFF substrate.

### 2.2. Characterization of the MgO@ACFF

Scan electron microscopy (SEM, JSM-7500F, Tokyo, Japan) and transmission electron microscopy (TEM, JEM-2010, Tokyo, Japan) were used to analyze the morphology of the MgO@ACFF. The structure and surface properties of the MgO@ACFF were characterized by X-ray diffraction patterns (XRD, D8 ADVANCE, New York, NY, USA), thermogravimetric (TG, STA409PC thermal analyzer, Bavaria, Germany), BrunauereEmmetteTeller (BET, Coulter Omnisorp 100CX, Brea, CA, USA), and X-ray photoelectron spectroscopy (XPS, Thermo ESCALAB 250Xi, Waltham, MA, USA), respectively.

### 2.3. Adsorption Experiments

The fluoride stock solution was prepared by de-ion water and NaF. The fluoride adsorption isotherm of the samples was obtained from the batch adsorption tests. Typically, a 300 mL conical flask with 100 mL of fluorine solution and 20 mg of the adsorbent was placed in a shaker at 150 rpm for 12 h. Then, the supernatant solution was pipetted and centrifuged for the determination of the remaining concentrations of fluoride.

In the kinetics study, the initial concentration of fluoride was 100 mg/L, and the adsorbent dose was 1.0 g/L. The flask was placed on a shaker for stirring at 150 rpm. At predetermined time intervals, stirring was interrupted, and 6 mL of supernatant solutions were pipetted and centrifuged for the determination of the remaining concentrations of fluoride.

In the study of the pH effect on fluoride removal, the pH value of the solutions was adjusted to a certain value using 0.1 mol HCl or NaOH solution. In the test of the influence of co-existing anions, chloride, nitrate, sulfate, bicarbonate, carbonate, and phosphate with different dosages of 0, 50, 100, 200, and 300 mg/g were added in the test flask. The mixed suspensions were also shaken at 150 rpm for 12 h. The adsorption capacity q_e_ (mg/g) was calculated from the following equation:(2)$$q_{e}=\frac{(C_{0}-C_{e})V}{m}$$
where C_0_ (mg/L) and C_e_ (mg/L) are the initial and equilibrium concentrations of fluoride, respectively, V (L) is the solution volume, and m (g) is the mass of the adsorbent.

## 3. Results and Discussion

### 3.1. Characterization of Materials

Figure 1 displays the XRD patterns of the pure ACFF, precursor-modified ACFF (precursor@ACFF), and MgO@ACFF. Obviously, only two broad amorphous packages can be seen from the XRD pattern of the ACFF. After modification, the peaks of the magnesium carbonate hydroxide hydrate (Mg_5_(CO_3_)_4_OH•4H_2_O, JCPDS No. 70-1177) appear in the diffraction patterns, indicating that the precursor has been successfully modified on the ACFF. In order to obtain the final product, the as-prepared sample is annealed in an N_2_ atmosphere at 500 °C for 2 h. As seen after annealing, the peaks of Mg_5_(CO_3_)_4_OH•4H_2_O vanish, and two obvious peaks at 42.9 and 62.3 emerge, which can be associated with the (200), and (220) planes of the hexagonal structure of MgO (JCPDS No. 89-7746). No precursor peak can be found after annealing, suggesting that the precursor is totally transformed into the final MgO products.

The morphologies of the pure ACFF and MgO@ACFF are studied by SEM. Figure 2a,b are the low and high-magnification SEM images of the pure ACFF. It can be seen that the ACFF consists of a huge number of active carbon fibers. The surface of each active carbon fiber is smooth, the diameter of which is about 10 μm on average. Figure 2c,d shows the SEM images of the ACFF after modification with different magnifications. Obviously, the diameter of the fiber increases a lot after modification. Figure 2e is the enlarged cross-section SEM image of the individual fiber after modification, from which a core-shell structure with the core of active carbon fiber and the shell of MgO can be obviously observed. The thickness of the MgO shell is about 100 μm. Figure 2f,g shows the cross-section SEM images of the MgO@ACFF. Apparently, the modified MgO nanomaterials have a lamellar structure, and a large number of MgO nanosheets are densely arranged on the surface of the ACFF. Figure 2h shows the EDS spectrum of the MgO@ACFF, the molar ratio of Mg:O is 1.87:1, which is close to the stoichiometry of MgO. Moreover, Mg and O, the peak of C can also be identified, which would be attributed to the conductive adhesive substrate. Figure 2i is the digital image of the MgO@ACFF. It is well known that ACFF is black; however, after modification, the color of the sample turns white, indicating the uniform modification of the MgO nanosheet on the ACFF substrate. Furthermore, the large size of the MgO@ACFF benefits the macro preparation and subsequent application of the product.

EDS mapping is used to further analyze the structure of the MgO@ACFF. Figure 3a is the SEM image of the individual porous MgO nanosheet-modified activated carbon fiber, and Figure 3b–d is the corresponding element mapping images of Mg, O, and C. It is clear that the shapes of the Mg and O elements are similar to the corresponding SEM image. From Figure 3d, it can be seen that, in the area corresponding to the shape of the fiber, the signal of C is very weak, implying the thick modification layer of porous MgO nanosheet. The rest red highlighted area would be attributed to the conductive adhesive. These results confirm that the ACFF is totally modified by a large number of MgO nanosheets.

Figure 4a,b is the TEM images of the porous MgO nanosheet and the precursor. The length of those nanosheets is about 1 μm, while the width is about 200 nm. The surface of the precursor nanosheet is very smooth. After annealing, the surface of the nanosheet becomes rough, and a large number of nanoholes emerge in the MgO nanosheet. It is believed that these nanoholes are caused by the condensation of the vacant part after the volatilization release of CO_2_ and H_2_O. Figure 4c shows the HRTEM image of modified MgO nanoplates, where the lattice spacing of 0.24 nm corresponds to the (111) plane of the hexagonal phase of MgO. The corresponding selected area electron diffraction (SAED) pattern (shown in Figure 4d) displays the shape of concentric circles, indicating the polycrystalline structure of the modified porous MgO nanosheet.

In order to study the thermal stability of the samples, the TG analysis of the sample is presented in Figure 5. When the temperature reaches 150 °C, the first stage of mass loss appears, as shown in Equation (3). The crystal water in the precursor escapes from the material as water vapor at high temperatures. With the increase in temperature, a second obvious stage of mass loss appears when the temperature reaches 350 °C, as shown in Equation (4), the precursor begins to decompose, and structural water and CO_2_ escape from the material. When the temperature is higher than 500 °C, the sample is quite stable. Thus, the annealing temperature is set at 500 °C.
Mg_5_(CO_3_)_4_(OH)_2_·4H_2_O → Mg_5_(CO_3_)_4_(OH)_2_ + 4 H_2_O↑(3)
Mg_5_(CO_3_)_4_(OH)_2_ → 5 MgO + 4 CO_2_↑ + H_2_O↑(4)

It should be noted that during the synthesis process of the MgO@ACFF, the pretreatment of the ACFF substrate is very important. The loading amount of MgO nanosheets can be greatly influenced by the pretreatment condition of the ACFF substrate. Figure 6a–d shows the SEM images of the MgO@ACFF samples with the ACFF substrates treated with different concentrations of NaOH solutions. When the ACFF substrates are treated with 0.1 M NaOH solution, the corresponding product is presented in Figure 6a. It can be seen that the modification of MgO is sparse and uneven. When the concentration of NaOH pretreatment solution increases to 0.5 mol/L, as shown in Figure 6b, the loading amount of MgO nanosheets on ACFF largely increases; consequently, the MgO nanosheets are intermittently attached to the surface of carbon fiber, leaving the surface of carbon fibers partially exposed. When the concentration of NaOH pretreatment solution increases to 1 mol/L, as shown in Figure 6c, the loading amount of MgO nanosheets on ACFF continuously increases until it totally covers the surface of the ACFF. The diameter of the modified fiber greatly increases as well. In contrast, when the concentration of NaOH pretreatment solution reaches 2 mol/L, parts of carbon fibers are not effectively wrapped by MgO nanosheets, and the bare surface can be observed (Figure 6d). Accordingly, the amount of modification of MgO nanosheets is reduced. Figure 6e presents the digital image of the MgO@ACFF samples with the ACFF substrates treated by NaOH solution with different concentrations. It can be seen that as the concentration of treated NaOH solution increases, the color of the MgO@ACFF becomes whiter. However, when the concentration of NaOH pretreatment solution is beyond 1 mol/L, an opposite trend is captured. Figure 6f summarizes the corresponding loading amount of the MgO nanosheet on the ACFF under different pretreatments. When the concentrations of NaOH pretreatment solution are 0.1, 0.5, 1, and 2 mol/L, the corresponding loading amounts of MgO nanosheets on ACFF are 220%, 395%, 810%, and 610%, respectively. Appendix A illustrates the Zeta potential of the ACFF treated with different NaOH solutions. Obviously, when ACFF is treated within 1 M NaOH solution, its zeta potential is more negative. The ACFF with a large negative zeta potential can adsorb more Mg^2+^ ions, thus having a higher loading amount [16,17].

To study the morphological evolution of the MgO@ACFF, the morphologies of the samples with different growth times are studied, and the corresponding SEM images are performed in Figure 7. Before the reaction, the surface of the active carbon fiber is smooth. However, after two hours, as seen in Figure 7a, the surface of the fiber becomes rough, and some small particles can be observed. When the reaction time reaches 4 h, it can be seen from Figure 7b that the activated carbon fiber is covered with a coarse and porous layer, and the diameter of the fiber grows to 20 μm. By extending the reaction time to 8 h, as present in Figure 7c, the layer of MgO continues to thicken, and the diameter of the fiber grows to almost 30 μm. On the surface of the fiber, some small nanosheets can be faintly seen. Finally, when the reaction time reaches 12 h (Figure 7d), those small pieces grow and, eventually, fully cover the surface of the fiber, and the porous MgO nanosheet-modified ACFF is finally formed. Appendix A presents the digital images of the MgO-modified ACFF with different reaction times. As reaction time goes on, more and more white products can be seen on the ACFF substrate. Figure 7e summarizes the variation of the loading amount of porous MgO nanosheets with the reaction time. It can be seen that the loading amount of porous MgO nanosheets continuously increases with the reaction time and finally reaches the equilibrium at 12 h. Therefore, to ensure the maximum loading amount of porous MgO nanosheets, the reaction time of our case was set at 12 h. Table 1 summarizes the loading amount of MgO nanomaterials on activated carbon in reported papers; obviously, the loading amount of MgO, in this case, is ten times higher than other reported results.

Based on the above results and analysis, the modification process of porous MgO nanosheets on ACFF can be deduced, and the corresponding schematic illustration is shown in Figure 8. Before reaction, ACFF is firstly treated with NaOH solution. In this way, the amount of surface hydroxyl on ACFF significantly increases. Then, the treated ACFF is put into the mixed aqueous solution of urea and magnesium sulfate at 120 °C. Due to a large amount of surface hydroxyl and a negative charge, a huge number of Mg^2+^ ions can be attracted to the surface of ACFF through static electricity. As urea decomposes at high temperatures, those Mg^2+^ ions on the ACFF surface could efficiently react with carbonate and hydroxide produced by decomposition, and basic magnesium carbonate nanoparticles are formed on the surface of ACFF. As the reaction goes by, with these nanoparticles as the nucleus, more and more basic magnesium carbonate products grow on the surface of ACFF. The basic magnesium carbonate products steadily grow and overlap each other and wrap on the surface of ACFF. After heat treatment, the precursor nanosheets are totally transformed into MgO nanosheets. During heat treatment, some gases, such as H_2_O and CO_2_, are released, while numerous vacancies are left. The condensation of vacancies results in the formation of a porous structure. In this way, the porous MgO nanosheet-modified ACFF is formed.

BET analysis is also used to analyze the microstructure of the MgO@ACFF. In contrast, the BET results of the ACFF and precursor@ACFF are also present in Figure 9. Figure 9a–c shows the N_2_ adsorption–desorption isotherms of the ACFF, precursor@ACFF, and MgO@ACFF, respectively. The N_2_ adsorption–desorption isotherm of ACFF belongs to the H2 hysteresis loop adsorption isotherm, indicating that the pore size distribution on its surface is uniform. The N_2_ adsorption–desorption isotherms of precursor@ACFF and MgO@ACFF both exhibit type IV isotherm with an H3 hysteresis loop, which reflects the mesoporous structures of the samples. Figure 9d shows the pore size distributions of the ACFF, precursor@ACFF, and MgO@ACFF, respectively. It can be found that after heat treatment, the amount of mesopores of the MgO@ACFF is significantly higher than that of the precursor@ACFF. The average pore diameter of the MgO@ACFF is 3.83 nm, which is consistent with the TEM result. After modification, the specific surface area of the ACFF and precursor@ACFF are 618.37 and 39.24 m^2^/g; this quick reduction indicates that the fiber surface is wrapped by the precursor. After heat treatment, the product of MgO@ACFF is formed, and the corresponding specific surface area increases to 134.83 m^2^/g, which can be attributed to the formation of a great amount of mesopores during the decomposition. The high BET surface area porous structure of the MgO@ACFF endows it with excellent adsorption performance.

### 3.2. Adsorption Properties

Firstly, the fluoride adsorption isotherms of the MgO@ACFF samples using the substrates with different pretreatment conditions are investigated, and the results are shown in Figure 10a. Obviously, the fluoride adsorption capacities of all MgO@ACFF samples are higher than that of the commercial MgO adsorbent. The fluoride adsorption property of pure ACFF also has been studied (Appendix A). As shown, the fluoride adsorption capacity of pure ACFF is less than 5 mg/g. Thus, the high adsorption performance of the MgO@ACFF is mainly attributable to the modified porous MgO nanosheets. As seen from Figure 10a, in the range of 0.1–1 mol/L, when the concentration of NaOH pretreatment solution increases, the fluoride adsorption performance of the corresponding MgO@ACFF samples increases; however, when the concentration of NaOH pretreatment solution is larger than 2 mol/L, the adsorption performance of the corresponding MgO@ACFF samples fall. As discussed earlier, different pretreatment conditions of the ACFF substrates directly lead to different loading amount of the porous MgO nanosheets. When the ACFF substrate is treated with 1 mol/L NaOH solution, the composite has the largest porous MgO nanosheet loading amount; thus, the corresponding sample exhibits the largest fluoride adsorption performance, and the fluoride adsorption capacity is more than 212.2 mg/g. Langmuir and Freundlich models, two common isotherm adsorption models, are applied to analyze those data, and the results are shown in Figure 10b,c. Table 2 summarizes the corresponding calculated isotherm parameters. The higher R^2^ values indicate that the adsorption isotherms of the MgO@ACFF absorbents follow the Freundlich model. The fluoride adsorption capacity of the MgO@ACFF also compares with formerly reported fluoride adsorbents, and the results are listed in Table 3. It can be seen that the fluoride adsorption capacity of the MgO@ACFF is not lower than that of pure nano-adsorbents and is much higher than that of loaded adsorbents.

The fluoride adsorption kinetics of the MgO@ACFF is further studied, as shown in Figure 11a. For comparison, the adsorption kinetics of commercial MgO adsorbent and pure ACFF is also studied. Clearly, in the first 50 min, the adsorption rate of fluoride on the MgO@ACFF is rapid, and almost 82% of the fluoride ions can be adsorbed. Then, the adsorption rate gradually slows down until reaching equilibrium. At adsorption equilibrium, more than 95% of the fluoride ions can be adsorbed by the MgO@ACFF. In contrast, only 47% and 2% of the fluoride ions can be removed by the commercial MgO adsorbent and pure ACFF. The experimental data of adsorption kinetic fit into the pseudo-first-order model as presented in Figure 11b. The regression coefficient (R^2^) of 0.999 indicates that the fluoride adsorption kinetics of the MgO@ACFF fitting favors the pseudo-second-order adsorption process.

The pH value is one of the most critical influential factors affecting the performance of fluoride adsorbents. Figure 11c shows the adsorption performance of MgO@ACFF to fluoride ions at different pH values. It can be seen that, in the pH range of 2.0–10.0, the MgO@ACFF exhibits an excellent fluoride adsorption capacity. However, when the pH value is larger than 11, the adsorption capacity of the MgO@ACFF significantly weakens. The reduction of the adsorption capacity in the alkaline condition can be attributed to the adsorption competition of hydroxyl ions with fluoride ions [33]. In the actual condition, various anions always coexist with fluoride ions. Therefore, it is necessary to study the influence of coexisting ions on the fluoride removal performance of MgO@ACFF. Figure 11d presents that, in the range of 0–300 mg/L, SO_4_^2−^, NO_3_^−^, and Cl^−^ have little effect on the fluoride adsorption capacity of the MgO@ACFF. However, HCO_3_^−^, CO_3_^2−^, and PO_4_^3−^ ions can greatly affect the fluoride removal performance of the MgO@ACFF. It is believed that the effect of these anions on fluoride removal performance is present because of the electrostatic interaction between those coexisting ions and the adsorbent. Those coexisting ions compete with fluoride ions for adsorption sites, thus causing performance degradation.

In order to illustrate the adsorption mechanism, the FTIR spectra of the MgO@ACFF before and after fluoride adsorption are presented in Figure 12. For MgO@ACFF, the FTIR peak at 431.07 cm^−1^ can be clearly seen, which corresponds to the Mg–O stretching vibration. After fluoride adsorption, the FTIR peak at 431.07 cm^−1^ vanishes, and a new FTIR peak at 503.63 cm^−1^ forms, which corresponds to the Mg–F stretching vibration [34], indicating the formation of the MgF_2_ after adsorption. The MgO@ACFF has a small FTIR peak at 3695 cm^−1^; this peak can be attributed to the stretching vibration modes of OH bands [30], suggesting the formation of Mg(OH)_2_ on the surface of MgO@ACFF. In addition, the FTIR peak of MgO@ACFF at 1444 cm^−1^, which corresponds to the asymmetric stretching vibration of carbonates, is greatly reduced after fluoride adsorption [35]. As a basic oxide, MgO reacts with CO_2_ molecules in the air and forms magnesium carbonate species on its surface. The reduction of this FTIR peak indicates that those surface carbonates are also involved in fluoride adsorption.

XPS analysis is employed to further reveal the adsorption mechanism of the MgO@ACFF, and the results are presented in Figure 13. From the survey spectra presented in Figure 13a, it can be seen that two more peaks located at 830.8 and 685.7 eV emerge after fluoride adsorption, which corresponds to F KLL and F1s [36], respectively. Figure 13b is the F1 spectra of MgO@ACFF before and after adsorption. Obviously, there is no peak before adsorption; however, after adsorption, a sharp peak emerges, indicating that a large amount of fluoride exists on the surface of the MgO@ACFF. Figure 13c presents the C1s spectra of MgO@ACFF before and after adsorption. Before adsorption, the C1s spectrum of the MgO@ACFF can be divided into three peaks of 284.8, 285.7, and 289.7 eV, which correspond to the ACFF substrate, surface carbonates [37], and adsorbed CO_2_ [11], respectively. After adsorption, the peak of 289.7eV is greatly weakened, while the peaks of 284.8 and 285.7 eV remain. This result indicates that the surface carbonates are consumed during the adsorption process. Figure 13d is the O1s spectra of the MgO@ACFF before and after adsorption. Before adsorption, two peaks located at 529.5 eV and 531.7 eV can be detected, which correspond to the MgO [38] and CO_3_^2−^ [39], respectively. After adsorption, the peak of 531.7 eV is largely weakened, while the peak of 529.5 eV vanished. Instead, a new peak of 531.2 eV assigned to the hydroxyl groups of Mg(OH)_2_ appears [40]. This result also indicates the replacement of the carbonates and the formation of Mg(OH)_2_. Both the FTIR and XPS results suggest the hydroxyl and carbonate co-exchange mechanism of the MgO@ACFF, and Figure 14 illustrates the adsorption mechanism of fluoride on the MgO@ACFF. During the adsorption process, the fluoride ions can be exchanged with the surface carbonates and hydroxyls and finally anchored on the MgO surface. Due to the large loading amount of the porous MgO nanosheets on the ACFF substrate, the fluoride removal performance of the MgO@ACFF is excellent. Furthermore, the enormous impact of the co-existed OH^−^ and CO_3_^2−^ on the adsorption performance further proves the co-exchange mechanism of the MgO@ACFF.

To further discuss the adsorption performance of MgO@ACFF, a breakthrough test has been carried out (Figure 15). When the concentration of fluoride in drinking water exceeds 1.0 mg/L, it would be harmful to human health. With an influent fluoride concentration of 5 mg/L, SV of 6 h^−1^ and pH of 6.8, the effluent fluoride was below 1 mg/L until the 505-bed volumes (BV) were treated by the MgO@ACFF. These results show that MgO@ACFF can be used in a packed bed for fluoride removal, especially in a rapid recycling and coupling process of adsorption and regeneration.

## 4. Conclusions

In conclusion, the porous MgO nanosheets were successfully modified on the surface of ACFF through a facile hydrothermal method. The MgO@ACFF presented a typical core-shell structure, and a great many porous MgO nanosheets were densely anchored on the surface of the activated carbon fiber felt. The loading amount of the porous MgO nanosheets is 810%, which is the largest reported loading amount concerned. The production process of the MgO@ACFF is quite simple, and the macro preparation of the MgO@ACFF can be realized. The fluoride adsorption rate of the MgO@ACFF is very high, which can be well fitted into a pseudo-second-order kinetic model. The adsorption rate of fluoride on the MgO@ACFF is rapid, and almost 82% of the fluoride ions can be adsorbed in 50 min. Meanwhile, the adsorption isotherm of the MgO@ACFF can be described by the Freundlich model, and the adsorption capacity is larger than 212.2 mg/g at a neutral state. The absorbent also showed high fluoride adsorption performance in a wide pH range of 2–10. The influence of co-existing ions also has been studied. The FTIR and XPS results revealed a hydroxyl and carbonate co-exchange mechanism. Furthermore, the column test indicates that 505-bed volumes of the 5 mg/L fluoride solution can be treated under 1.0 mg/L. This work implies that the MgO@ACFF is a potentially suitable candidate for fluoride removal.

## Figures and Tables

**Figure 1 nanomaterials-13-01082-f001:**
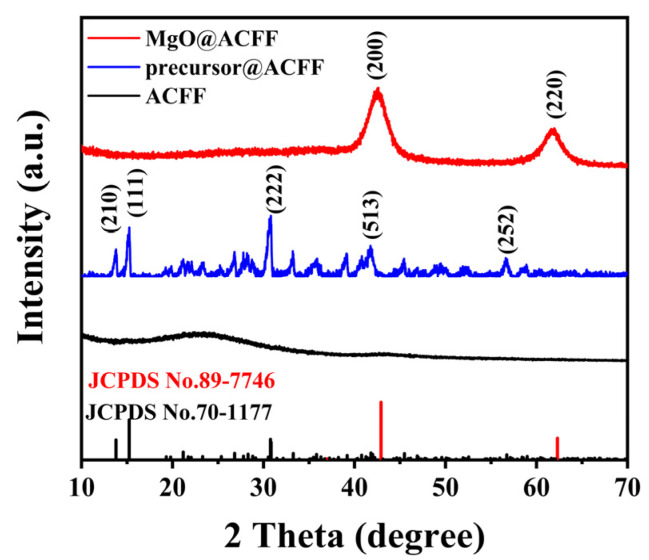
XRD patterns of the ACFF, precursor@ACFF, and MgO@ACFF.

**Figure 2 nanomaterials-13-01082-f002:**
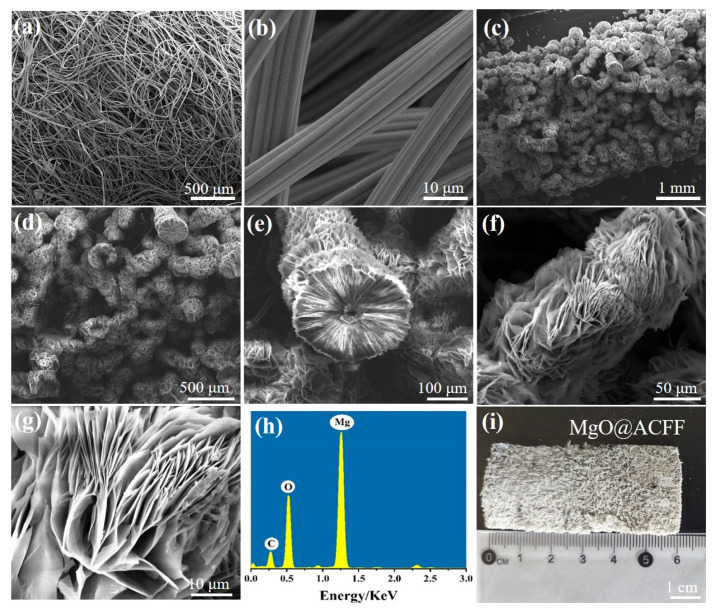
(**a**,**b**) low and high magnification SEM images of the ACFF; (**c**–**e**) SEM images of the MgO@ACFF with different magnification; (**f**,**g**) low and high magnification side SEM images of the MgO@ACFF; (**h**) EDS spectrum of the MgO@ACFF; (**i**) digital image of the MgO@ACFF.

**Figure 3 nanomaterials-13-01082-f003:**
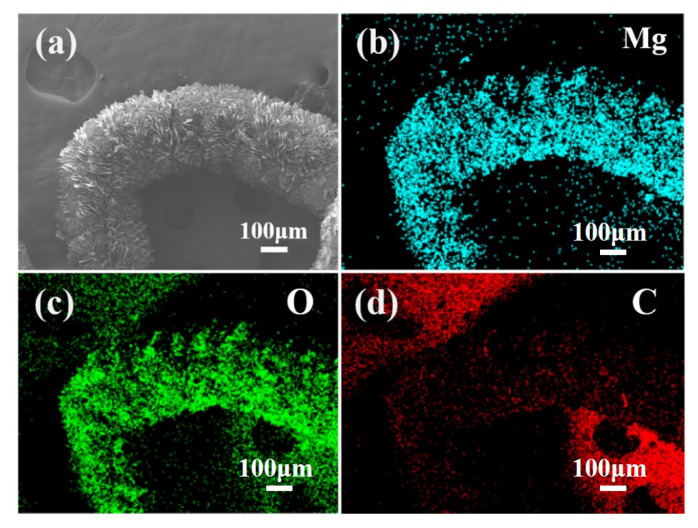
(**a**) SEM image of the MgO@ACFF, (**b**–**d**) the corresponding element mapping images of the MgO@ACFF.

**Figure 4 nanomaterials-13-01082-f004:**
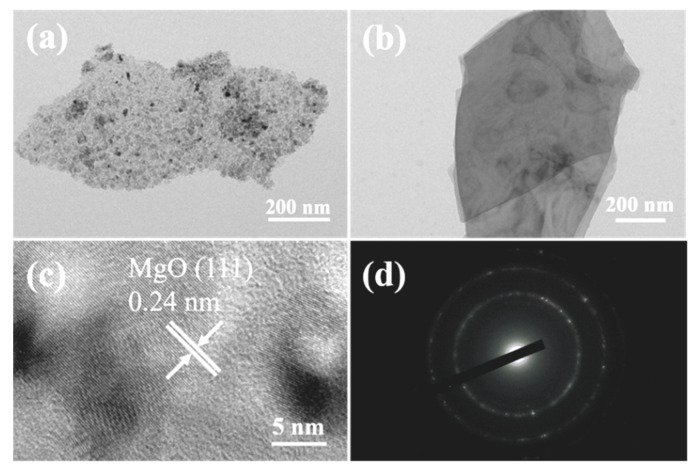
(**a**,**b**) The TEM images of the modified MgO nanosheets and precursor; (**c**) shows the HRTEM image of modified MgO nanosheets; and (**d**) the corresponding SAED pattern.

**Figure 5 nanomaterials-13-01082-f005:**
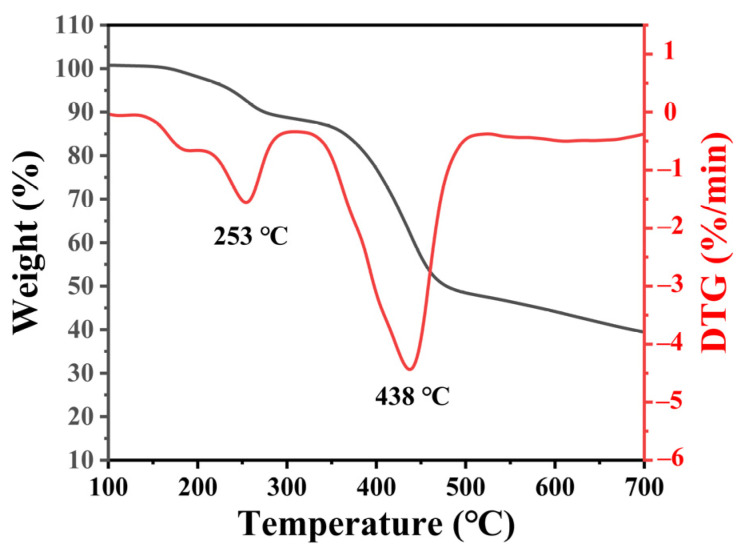
TG curve of the precursor@ACFF.

**Figure 6 nanomaterials-13-01082-f006:**
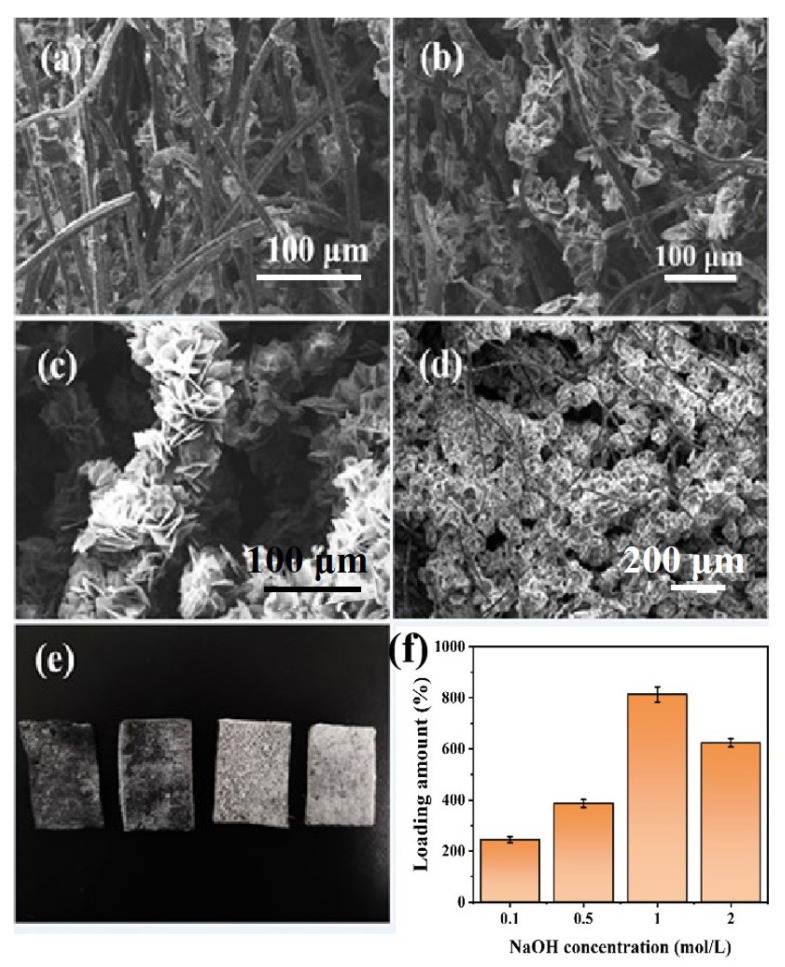
(**a**–**d**) SEM images of the MgO@ACFF samples with the ACFF substrates treated by 0.1, 0.5, 1, and 2 mol/L NaOH solutions; (**e**) digital image of the corresponding samples; (**f**) the corresponding loading amount of MgO nanosheet on the ACFF under different pretreatments.

**Figure 7 nanomaterials-13-01082-f007:**
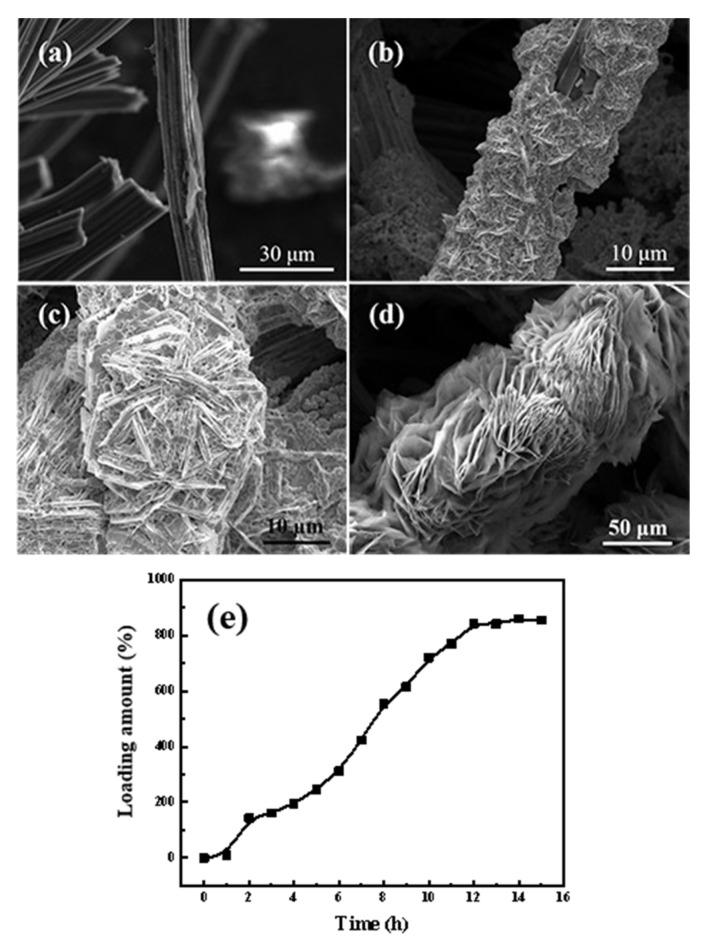
(**a**–**d**) SEM images of the MgO@ACFF at different growth times (2, 4, 8, 12 h, respectively); (**e**) the variation of loading amount of porous MgO nanosheets with reaction time.

**Figure 8 nanomaterials-13-01082-f008:**
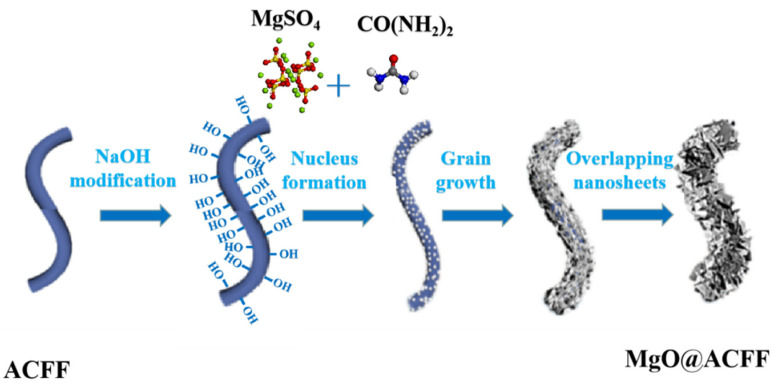
Schematic illustration of the formation of the MgO@ACFF.

**Figure 9 nanomaterials-13-01082-f009:**
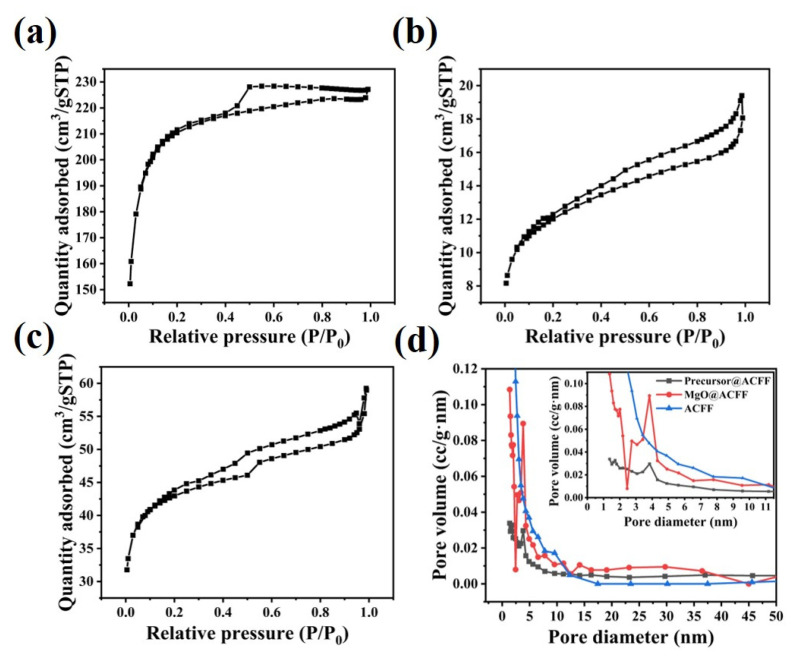
(**a**–**c**) the N_2_ adsorption–desorption isotherm of the ACFF, precursor@ACFF, and MgO@ACFF. (**d**) the corresponding pore size distributions of the ACFF, precursor@ACFF, and MgO@ACFF.

**Figure 10 nanomaterials-13-01082-f010:**
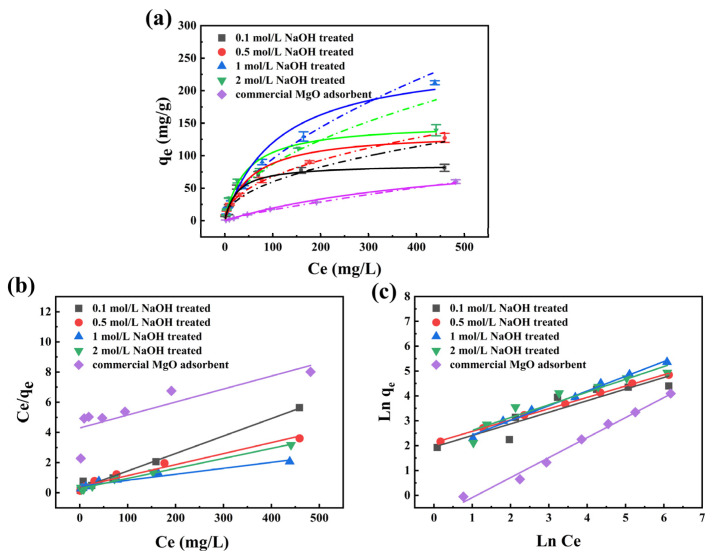
(**a**) Adsorption isotherm of fluoride on the MgO@ACFF with the substrate treated with different concentrations NaOH solutions; (**b**) the corresponding linear fitting of Langmuir model; (**c**) the corresponding linear fitting of Freundlich model (0.3 g/L adsorbent, pH = 7.0, 35 °C).

**Figure 11 nanomaterials-13-01082-f011:**
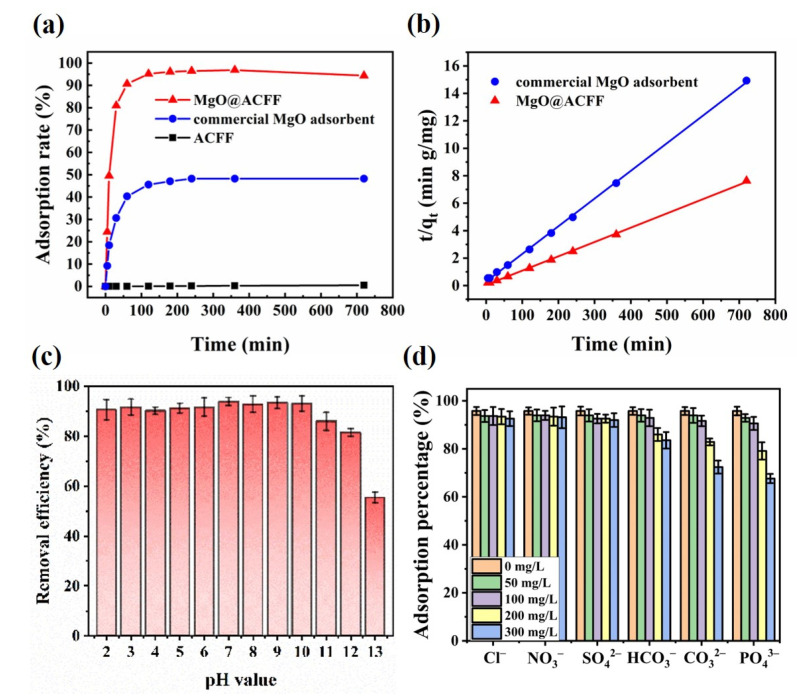
(**a**) adsorption kinetics of fluoride removal by the MgO@ACFF; (**b**) the corresponding pseudo-second order fitting; (**c**) adsorption performance of MgO@ACFF to fluoride ions at different pH values; (**d**) effect of the competing anions on the MgO@ACFF.

**Figure 12 nanomaterials-13-01082-f012:**
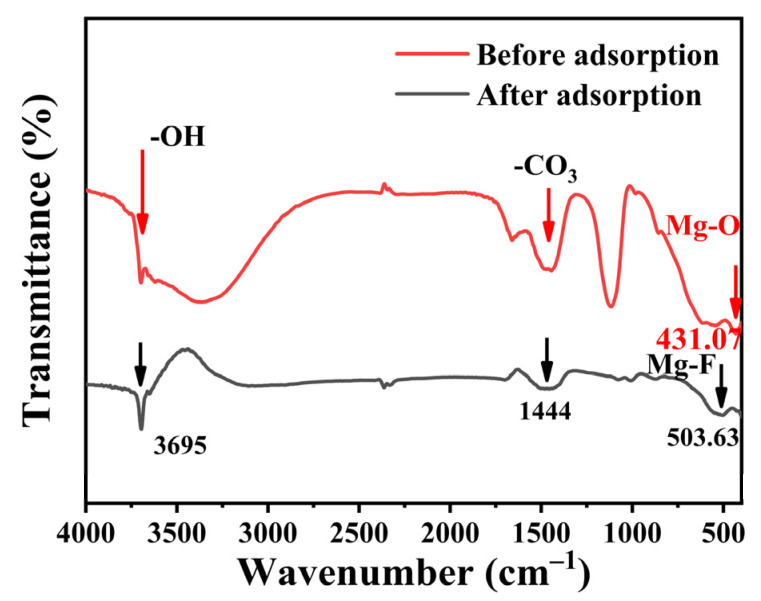
FTIR spectrum of the MgO@ACFF before and after adsorption.

**Figure 13 nanomaterials-13-01082-f013:**
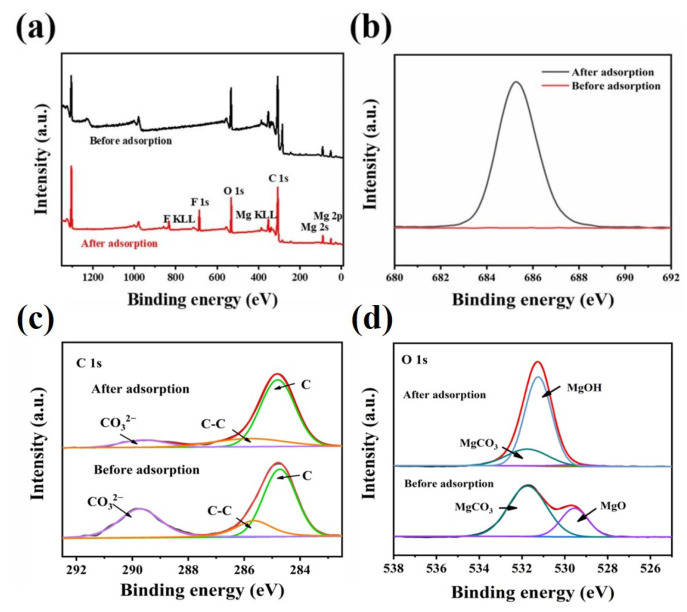
(**a**) survey spectra of MgO@ACFF before and after fluoride adsorption; (**b**–**d**) F1s, C1s, and O1s spectra of the of MgO@ACFF before and after fluoride adsorption.

**Figure 14 nanomaterials-13-01082-f014:**
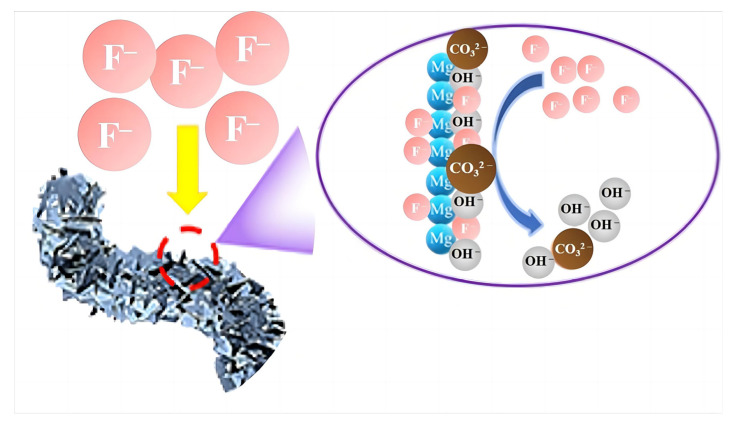
Adsorption mechanism of the MgO@ACFF.

**Figure 15 nanomaterials-13-01082-f015:**
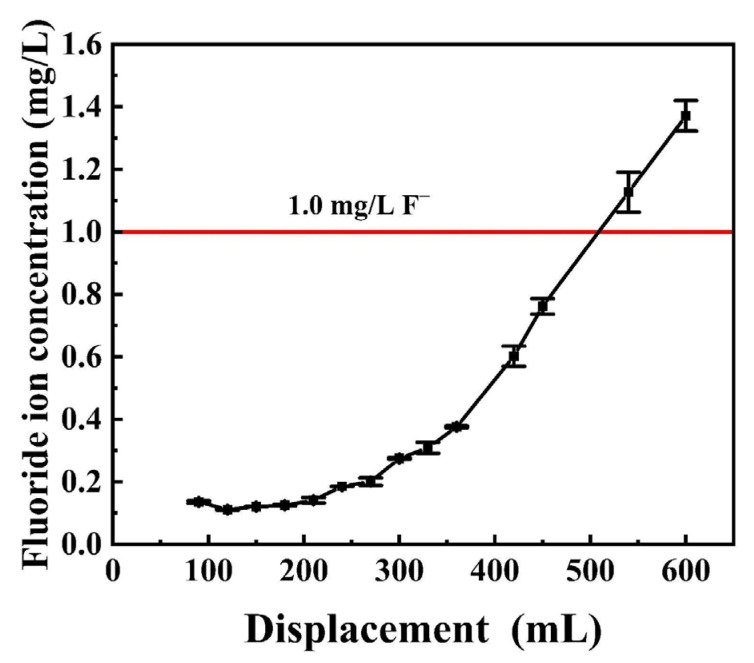
Breakthrough curve for fluoride adsorption of the MgO@ACFF.

**Table 1 nanomaterials-13-01082-t001:** The comparison of the MgO loading amount in MgO/activated carbon composites.

Material	Loading Amount (Weight%)
AC-Mg [18]	22.58
AC-Mg [19]	0.65
AC-Mn [19]	3.56
AC-Fe-U [20]	17.33 ± 0.71
AC-La [21]	12
PAC-TNTs [22]	70.94
AC-Zn/AC-Mn [23]	16.76/14.44
EV-La(OH)_3_ [24]	31.52
This work	810

**Table 2 nanomaterials-13-01082-t002:** Langmuir model and Freundlich fitting parameters of MgO@ACFF by activation of NOH at different concentrations.

Equations	Langmuir Model			Freundlich Model		
	qe=qmKLCⅇ1+KLCⅇ			qe=qmKLCⅇ1+KLCⅇ		
Viscosity	q_m_ (mg/g)	K_L_ (L/mg)	R^2^	K_F_ (mg/g)	1/n	R^2^
**0.1 M**	86.356	0.040	0.989	6.970	2.144	0.823
**0.5 M**	136.799	0.018	0.961	8.321	2.196	0.998
**1 M**	253.165	0.009	0.928	6.230	1.689	0.991
**2 M**	150.60	0.023	0.991	8.215	1.952	0.888
**Commercial MgO**	45.186	0.002	0.636	0.408	1.242	0.986

**Table 3 nanomaterials-13-01082-t003:** Fluoride adsorption capacities of different adsorbents.

Adsorbent	Adsorption Capacity (mg/g)	Dose (g/L)	pH
Alumina [25]	83.3	1	6
Aluminum hydroxide [26]	110	0.081	6
Al-modified hydroxyapatite [27]	32.57	5	7
La (III)–modified GAC [9]	9.96	0.1	7
AC-Zr [28]	5.4	2	5.44
AC-Mn-Mg-Zr [19]	26.67	1	11.9
Mg-Fe-La hydrotalcite-like compound [29]	59.98	0.5	6.8
orous hollow MgO microspheres [30]	>120	1	7
Mg–Al bimetallic oxides [31]	89.3	1	6
MgO/chitosan [32]	>4.44	2	7
porous MgO nanosheets [11]	>185.5	1	7
MgO@ACFF (This work)	>212.2	0.3	neutral

## Data Availability

The data presented in this study are openly available.

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
