# Peer review of "Synthesis and Characterization of Porous MgO Nanosheet-Modified Activated Carbon Fiber Felt for Fluoride Adsorption"

_nanomaterials, 2023, doi:10.3390/nano13061082_

Round 1
Reviewer 1 Report
This manuscript reports on the synthesis of activated carbon fiber felt as modified by porous MgO nanosheets (MgO@ACFF) considered for efficient fluoride removal (from water). Characterization was conducted by XRD, SEM, TEM, EDS and BET. Adsorption capacity of MgO@ACFF was then studied in detail considering co-existence of processes and effects. FTIR and XPS analysis was adequately employed too. This topic is not much studied, and the present work is timely. Also the results are valuable and appealing, and thus can be very helpful for the interested community to make choices and benchmarking when future attempts on modifying material systems for efficient adsorption (not only of fluoride). Useful and employable future developments and solutions in relation to this system can be extracted from the present results. Linguistically, in terms of writing clarity and expressions the manuscript needs improvement.
All in all, this work certainly represents a valuable contribution with possible wider impact in the field once stylistic, terminological and linguistic aspects of the manuscript are improved.
The authors chose an adequate structure of the manuscript. Also, they provided concise and nicely formatted figures and corresponding analysis which can attract significant scientific and practical interest and adds new knowledge to the field.
The present manuscript is a significant contribution, this work once published would be instructive and suggestive in terms of further studies and with good chances be cited.
There are some relatively minor issues with this already good manuscript that will need to be addressed before the manuscript becoming suitable for publication, i.e., it can be considered for publication after a minor to moderate revision:
1: Title should be significantly improved. Firstly, using adjectives like “facile” and even more “excellent” is not a good style in scientific writing besides that such wording is trivial in this context. Also, the phrase “MgO porous nanosheets modified activated carbon fiber felt composite” would not be understand by 95% of the readers. Authors may be want to say “activated carbon fiber felt composite modified by porous MgO nanosheets”. In addition, it is incorrect to say “MgO porous nanosheet”, correct word order is “porous MgO nanosheets” …. And so on and so forth. An expert more fluent in English should look at the entire manuscript!
2: Abstract should more explicitly mention the adsorption efficiency of the material system, something that becomes quite clear from the good results obtained. It is not well transmitted in the Abstract and to a lesser extent by the conclusions though.
3: The issue of the thermal stability of MgO@ACFF should be concretized or at least mentioned. Is there any quantitative data about the thermal stability of the material system? Empiric data?
5: In the introduction, the authors miss that similarly complex materials where modification based on nanostructure occurs can be assessed by systematic atomistic modeling which can guide the synthesis of the system and directly explain its properties [Dalton Transactions 44 (2015) 3356-3366, Chemical Physics Letters 458 (2008) 170-174]. Such works should be cited in the present manuscript to emphasize that theoretical support is available in such cases.
5: Spell-check and stylistic revision of the paper are very much necessary necessary. Some long sentences, wrong word order, unclear phrases, as well as misspellings, etc., are very much noticeable throughout the text.
Author Response
- Title should be significantly improved. Firstly, using adjectives like “facile” and even more “excellent” is not a good style in scientific writing besides that such wording is trivial in this context. Also, the phrase “MgO porous nanosheets modified activated carbon fiber felt composite” would not be understand by 95% of the readers. Authors may be want to say “activated carbon fiber felt composite modified by porous MgO nanosheets”. In addition, it is incorrect to say “MgO porous nanosheet”, correct word order is “porous MgO nanosheets” …. And so on and so forth. An expert more fluent in English should look at the entire manuscript!
=> Thanks very much for the reviewer’s comments. As the suggestion of the reviewer, l the word "MgO porous nanosheets" has changed to "porous MgO nanosheets". And, the title of the manuscript has changed to “Synthesis and characterization of porous MgO nanosheets modified activated carbon fiber felt for fluoride adsorption”. Furthermore, several workmates with high English standard have helped us to check the manuscript, and numerous mistakes have been corrected. All changes made to the text are in Red color.
- Abstract should more explicitly mention the adsorption efficiency of the material system, something that becomes quite clear from the good results obtained. It is not well transmitted in the Abstract and to a lesser extent by the conclusions though.
=> Thanks very much for the reviewer’s comments. The part of abstract has been totally rewritten, and the the adsorption performance of the material has been explicitly expressed
- The issue of the thermal stability of MgO@ACFF should be concretized or at least mentioned. Is there any quantitative data about the thermal stability of the material system? Empiric data?
=> Thanks very much for the reviewer’s comments. In order to In order to study the thermal stability of the samples, the TG analysis of the sample is carry out. As seen from Figure 5, when the heating temperature reaches 150℃, the first stage of mass loss appears. As present in Eq.(1), the crystal water in the precursor escapes from the material. With the increase of heating temperature, second stage of mass loss appears when the temperature reaches 350℃, as shown in Eq.(2), the precursor begins to decompose, and structural water and CO2 escape from the material. When the temperature larger than 500℃, with the end of decomposition reaction, Significant increase in the thermal stability significantly increase. Thus, the anneal temperature is set at 500℃.
Mg5(CO3)4(OH)2·4H2O → Mg5(CO3)4(OH)2+4 H2O ↑ (1)
Mg5(CO3)4(OH)2 → 5 MgO+4 CO2 ↑+ H2O ↑ (2)
Figure R1. TG curve of the precursor@ACFF
- In the introduction, the authors miss that similarly complex materials where modification based on nanostructure occurs can be assessed by systematic atomistic modeling which can guide the synthesis of the system and directly explain its properties [Dalton Transactions 44 (2015) 3356-3366, Chemical Physics Letters 458 (2008) 170-174]. Such works should be cited in the present manuscript to emphasize that theoretical support is available in such cases.
=>Thanks very much for the reviewer’s comments. We have cited those papers in the revised manuscript.
- Spell-check and stylistic revision of the paper are very much necessary necessary. Some long sentences, wrong word order, unclear phrases, as well as misspellings, etc., are very much noticeable throughout the text.
=>Thanks very much for the reviewer’s comments. Several workmates with high English standard have helped us to check the manuscript, and numerous mistakes have been corrected in the revised manuscript.

Reviewer 2 Report
The manuscript reports the development of a new adsorbent called MgO@ACFF for efficient fluoride removal from water. The adsorbent has a core-shell structure with MgO porous nanosheets densely anchored on the surface of activated carbon fiber felt. The adsorption kinetics of MgO@ACFF follows a pseudo-second-order kinetic model, and its adsorption capacity is higher than 212.2 mg/g at neutral pH. The presence of co-existing ions has little effect on the adsorption capacity of MgO@ACFF, and its adsorption mechanism involves a hydroxyl and carbonate co-exchange mechanism. The column test confirms the potential of MgO@ACFF for fluoride removal. The unique structure of the adsorbent provides high surface area and active sites for fluoride adsorption, making it a promising candidate for practical applications.
I think this paper contributes to the field of environmental engineering, particularly in the area of water treatment and pollutant removal. Overall, this manuscript represents a contribution to the field and is recommended for publication in nanomaterials with the following revisions. Prior to publication, authors should address the following minor issues listed below:
1. It would be helpful to include the pXRD of MgO as a reference in Figure 1.
2. In Figure 2, the x-coordinate of Figure 2h appears to have low resolution and is difficult to see clearly. Additionally, the abscissa of Figure 2 should be labeled with a name and units. Lastly, it would be beneficial to maintain the same thickness and position of the scale bar across Figures 2a-g. The same concerns regarding the scale bars are present in the other figures, and it would be beneficial to maintain consistency across all figures.
3. In Figure 2h and Figure 3, the authors state that ACFF is fully enclosed by a large number of MgO nanosheets. However, EDS analysis can detect elements even when they are covered by micron-sized materials. Additionally, the obvious peak of ACFF is visible in the pXRD, and the strong C peak is apparent in the XPS, which conflicts with the SEM-EDS data. Therefore, the authors should provide additional clarification on this matter.
4. Supplementary Materials are missing for review.
5. It is suggested to include the most recent literature on activated carbon fiber felt (ACFF) to support and strengthen the findings of this study.
a. Nano Research Energy 2022, 1: e9120022 (High-efficiency electrocatalytic NO reduction to NH3 by nanoporous VN)
b. Nanomaterials 2022, 12(3), 427 (Enhancing the Performance of a Metal-Free Self-Supported Carbon Felt-Based Supercapacitor with Facile Two-Step Electrochemical Activation)
Author Response
Response to Reviewer
- It would be helpful to include the pXRD of MgO as a reference in Figure 1.
=>Thanks very much for the reviewer’s comments. As shown in Figure R1, the MgO diffraction peak position in the standard PDF card was added in the XRD pattern.
Figure R1. XRD patterns of the ACFF, precursor@ACFF and MgO@ACFF.
- In Figure 2, the x-coordinate of Figure 2h appears to have low resolution and is difficult to see clearly. Additionally, the abscissa of Figure 2 should be labeled with a name and units. Lastly, it would be beneficial to maintain the same thickness and position of the scale bar across Figures 2a-g. The same concerns regarding the scale bars are present in the other figures, and it would be beneficial to maintain consistency across all figures.
=> Thanks very much for the reviewer’s comments. As present in Figure R2 (Figure 2 in the revised manuscript), Figure R2h (Figure 2h in the revised manuscript ) was replaced with a clearer one. The abscissa of Figure R2 (Figure 2 in the revised manuscript) was labeled with name and units. And, the scale bar in Figures R2 a-g (Figures 2a-g in the revised manuscript) was revised with the same thickness and position. Figure R3 (Figure 6 in the revised manuscript) also has been revised in the revised manuscript.
Figure R2. (a, b) low and high magnification SEM images of the ACFF, (c-e) SEM images of the MgO@ACFF with different magnification, (f, g) low and high magnification side SEM images of the MgO@ACFF, (h) EDS spectrum of the MgO@ACFF; (i) digital image of the MgO@ACFF.
Figure R3. (a-d) SEM images of the MgO@ACFF samples with the ACFF substrates treated by 0.1, 0.5, 1 and 2 mol/L NaOH solutions, (e) digital image of the corresponding samples, (f) the corresponding loading amount of MgO nanosheet on the ACFF under different pretreatments.
- In Figure 2h and Figure 3, the authors state that ACFF is fully enclosed by a large number of MgO nanosheets. However, EDS analysis can detect elements even when they are covered by micron-sized materials. Additionally, the obvious peak of ACFF is visible in the pXRD, and the strong C peak is apparent in the XPS, which conflicts with the SEM-EDS data. Therefore, the authors should provide additional clarification on this matter.
=>Thanks very much for the reviewer’s comments. The EDS analysis present in Figure 2h and Figure 3 are obtained from the scanning electron microscope, during sample preparation, the MgO@ACFF was fixed on the conductive adhesive layer. The carbon element in EDS analysis is attribute to the carbon in the conductive adhesive. The porous MgO nanosheets are densely arranged on the surface of the fiber, however, the thickness of the MgO nanosheets is quite thin, thus there are large pores between nanosheets, which can be clearly seen in Figure 2f and g. The arrangement direction of the nanosheets is generally parallel to the electron incident direction, thus some electrons can pass through the sample, and the carbon signal of the conductive adhesive layer can be detected. Notably, in the area corresponding to the shape of fiber, the signal of C is very weak, implying the thick modification layer of porous MgO nanosheet.
In the test of XRD, due to the limitation of the size, the MgO@ACFF samples need to cut to the right size. In the process of cutting, part of the modified porous MgO nanosheet may fall off from the fiber, and the surface of the activated carbon is exposed. And, for better explanation the existence of ACFF and porous MgO nanosheet, this sample is not the sample with the highest loading amount. In this way, both of the diffraction peaks of ACFF and porous MgO nanosheet can be clearly seen. As the suggestion of the reviewer, in the revised manuscript, the XRD patterns of the samples were replaced by Figure R4.
For the test of XPS, the MgO@ACFF samples need to cut to the right size. And, during the cutting, adsorption, separation and drying process, part of the modified porous MgO nanosheet may also fall off from the fiber. Thus, in XPS result, the carbon peak is obvious.
We have clarification on this in the revised manuscript.
Figure R4. XRD patterns of the ACFF, precursor@ACFF and MgO@ACFF.
- Supplementary Materials are missing for review.
=> Thanks very much for the reviewer’s comments. Supplementary Material was uploaded.
- It is suggested to include the most recent literature on activated carbon fiber felt (ACFF) to support and strengthen the findings of this study.Nano Research Energy 2022, 1: e9120022 (High-efficiency electrocatalytic NO reduction to NH3 by nanoporous VN)Nanomaterials 2022, 12(3), 427 (Enhancing the Performance of a Metal-Free Self-Supported Carbon Felt-Based Supercapacitor with Facile Two-Step Electrochemical Activation)
=>Thanks very much for the reviewer’s comments. Those papers have been cited in the revised manuscript.
